# Pyrroles as Privileged Scaffolds in the Search for New Potential HIV Inhibitors

**DOI:** 10.3390/ph14090893

**Published:** 2021-09-02

**Authors:** Maria da Conceição Avelino Dias Bianco, Debora Inacio Leite Firmino Marinho, Lucas Villas Boas Hoelz, Monica Macedo Bastos, Nubia Boechat

**Affiliations:** 1Laboratório de Sintese de Farmacos—LASFAR, Instituto de Tecnologia em Farmacos—Farmanguinhos, FIOCRUZ, Rua Sizenando Nabuco 100, Manguinhos, Rio de Janeiro 21041-250, RJ, Brazil; mariaconcz@hotmail.com (M.d.C.A.D.B.); debora.marinho@far.fiocruz.br (D.I.L.F.M.); lucas.hoelz@far.fiocruz.br (L.V.B.H.); monica.macedo@far.fiocruz.br (M.M.B.); 2Centro de Ciencias da Saude—CCS, Programa de Pos-graduação em Farmacologia e Quimica Medicinal do Instituto de Ciencias Biomedicas—ICB-UFRJ, Bloco J, Ilha do Fundao, Rio de Janeiro 21941-902, RJ, Brazil

**Keywords:** pyrrole, heterocycle, AIDS, HIV, CD4 antagonist, glycoprotein, entry-inhibitors, reverse transcriptase, integrase

## Abstract

Acquired immunodeficiency syndrome (AIDS) is caused by human immunodeficiency virus (HIV) and remains a global health problem four decades after the report of its first case. Despite success in viral load suppression and the increase in patient survival due to combined antiretroviral therapy (cART), the development of new drugs has become imperative due to strains that have become resistant to antiretrovirals. In this context, there has been a continuous search for new anti-HIV agents based on several chemical scaffolds, including nitrogenated heterocyclic pyrrole rings, which have been included in several compounds with antiretroviral activity. Thus, this review aims to describe pyrrole-based compounds with anti-HIV activity as a new potential treatment against AIDS, covering the period between 2015 and 2020. Our research allowed us to conclude that pyrrole derivatives are still worth exploring, as they may provide highly active compounds targeting different steps of the HIV-1 replication cycle and act with an innovative mechanism.

## 1. Introduction

Acquired immunodeficiency syndrome (AIDS) is caused by the human immunodeficiency virus (HIV) and remains a global health problem four decades after the report of its first case [1]. As of 2019, there were 38 million people living with HIV infection globally. Nonetheless, millions of lives have been saved through the combination of antiretroviral therapy (cART) access since 1996 [2].

cART, usually called an AIDS cocktail, is composed of more than one drug, and must be administered daily for the lifetime of the infected patient once HIV has been integrated into the host cell genome DNA. CD4+ Lymphocyte infection and destruction result in the severe immunosuppression features of AIDS, allowing the establishment of secondary infections [3,4]. The available anti-HIV drugs target the essential HIV enzymes reverse transcriptase (RT), integrase (IN), and protease. However, other stages of HIV infection can be addressed, for example, entry and capsid inhibitors [5,6]. There is no cure for this disease or a vaccine to help to avoid HIV transmission, and drug-based therapy is the only means to control infection progress [7].

Despite the success of viral load suppression and increased patient survival due to the correct administration of cART, the development of new drugs and formulations of combined fixed doses (CFDs), and long-releasing formulations with better patient adherence and tolerability to treatment, is imperative. These aspects are also essential for decreasing the selection of drug-resistant viral strains, which are currently one of the significant problems of therapy [6,8]. In addition, several drawbacks have been developed due to cART, including long-term toxicity and drug–drug interactions [9].

In this context, there has been a continuous search for new anti-HIV agents based on several chemical groups, including nitrogenated heterocycles such as pyrroles [8]. The application of heterocycles is a strategy that is widely used in medicinal chemistry to obtain drug candidates because it can modulate properties related to pharmacokinetics, such as solubility, lipophilicity, polarity, and pharmacodynamics, such as hydrogen-bonding capacity and the ability to form complexes with coordinating metals [10,11].

The pyrrole ring (Figure 1) is a five-membered aromatic heterocycle with electron-rich characteristics. It comprises four carbon atoms and was isolated and characterized for the first time in 1857 [12,13]. The pyrrole moiety is obtained through various synthetic methods widely discussed in the literature [14], besides being found extensively in natural products [11]. This heterocycle is commonly found in molecules with several biological activities, for example, in drugs approved by the Food and Drug Administration (FDA), such as the anticancer drug sunitinib (Figure 1). It is also possible to find this nucleus fused with other cycles in antiemetic, anti-inflammatory, and antiviral agents such as remdesivir (Figure 1) [10,14].

Researchers have summarized the applications of pyrrole as a privileged structure in the development of potential anti-HIV drug candidates [8,10,15,16,17,18,19,20], in particular targeting HIV-1 gp-120, RT, and IN. In this review, we provide an overview of pyrrole-based compounds with anti-HIV activity as possible future trends in the treatment of AIDS. We explored the literature between 2015 and 2020, selecting the most relevant papers that showed the importance of pyrroles and their structure-activity relationship (SAR) data as HIV inhibitors.

## 2. Pyrrole-Based Entry Inhibitors of the HIV-1

The beginning of the HIV replication cycle occurs through attachment between the CD4 cell receptor and the HIV surface envelope glycoprotein gp120. This event forces gp120 to change its conformation, facilitating binding to the host cell coreceptor CCR5 (in M-tropic viruses) or CXCR4 (in T-tropic viruses) in the V3 loop of gp120 [21]. The CD4 binding site of the HIV-1 gp120 envelope, termed the Phe43 cavity, is highly conserved and has been hypothesized to be a site less prone to resistance-conferring mutations [22]. The fusion step occurs with the assistance of gp41, which also undergoes a conformational change due to the interactions described earlier [23,24].

Therefore, based on this initial infection process, the inhibitors that block the attachment or fusion process are collectively termed “entry inhibitors” and can act on the glycoproteins (gp120 and gp41), and receptors and coreceptors present on the host cell surface (CD4 and CCR5 or CXCR4). Maraviroc and enfuvirtide are currently available CCR5 and gp41 inhibitors, respectively [24,25].

Fostemsavir (Rukobia), a prodrug of temsavir (Figure 2), is an entry inhibitor that was most recently approved in the United States by the FDA in July 2020 and Europe by the European Medicine Agency (EMA) in the same year [26,27]. This drug binds directly to the gp120 subunit surface-accessible pocket at the interface between the inner and outer domains of gp120 under the β20–21 loop interacting with the C terminus of the α1-helix [28]. This interaction process blocks HIV from attaching to the CD4+ T cells of the host immune system and prevents the virus from infecting these cells and multiplying [29,30]. Temsavir has been described to act against CCR5-, CXCR4-, and dual-tropic (R5X4) strains of HIV-1 [31,32,33,34,35]. It is essential to mention that fostemsavir is the first antiretroviral drug that contains a pyrrolopyridine ring in its chemical structure (Figure 2).

The introduction of a phosphate group at the *N*-position in the pyrrole ring allows fostemsavir to improve its solubility in the gut, facilitating efficient dissolution. Fostemsavir is converted to temsavir by alkaline phosphatase in the gastrointestinal lumen. Thus, the drug can rapidly be absorbed due to its efficient membrane permeability, and the NH group on the pyrrolopyridine ring is free to take part in the key interactions with the gp120 [36].

The crystal structure of the gp120-temsavir complex (PDB code: 5u7o) shows that temsavir binds to a surface-accessible cavity interface between the inner and outer domains of the glycoprotein under the β20–21 loop, interacting with the C terminus of the α1-helix (Figure 3A) [30]. According to Pancera and coworkers [30], temsavir interacts with gp120 mainly through steric interactions, including Ile108, Ile109, Trp112, Asp113, Leu116, Thr202, Val255, Ser256, Thr257, Glu370, Ser375, Phe376, Phe382, Phe383, Tyr384, Ile424, Asn425, Met426, Trp427, Asn428, Gln432, Ala433, Met434, and Met475 residues (Figure 3B,C). In addition, two hydrogen bonds are present in the temsavir-gp120 complex: one between the backbone NH of Trp427 and its oxoacetamide carbonyl, and the other between the side-chain carboxylate of Asp113 and the NH group of its pyrrolopyridine ring (Figure 3D). Other interactions include the benzoyl group of temsavir, which enables aromatic stacking with Phe382 and Trp427 of gp120 (Figure 3E).

Despite advances in the discovery and approval of new entry inhibitors, several research groups have been working to obtain molecules of this antiretroviral class [37,38,39,40,41,42,43]. One example was the discovery of NBD-11021 (**1**) (Figure 4) [37], which presented antagonistic entry properties for HIV-1 and was the precursor for the development of new derivatives with similar characteristics. The design of this molecule was based on changing the oxalamide in the precursor NBD-09027 (**2**) to a pyrrole ring (Figure 4). This change enabled the conversion of a compound that was a CD4 agonist and enhanced the entry of HIV-1 into the cell into a full CD4 antagonist (Figure 4). The pyrrole ring rigidity forces conformational changes, leading to the formation of a hydrogen bond between the piperidine ring and Asp368 in the cavity of gp120 [37]. Prototype **2** does not present this kind of interaction. However, the authors explain it is not possible to attribute the CD4 antagonism from this interaction due to the restricted X-ray structural information.

Compound **1** yielded a diastereomeric mixture (four isomers), which authors designated “NBD-11021” and used in different assays [37].

Compound **1** was more active than **2** in both single- and multicycle assay (Table 1) [37]. Another important discovery was the capacity of **1** to inhibit both CCR5- and CXCR4-tropic HIV-1 with similar potency (IC_50_ of ~1.7–2.4 μM) [37].

The gp120 crystal structure core in the complex with **1** indicates that a cumulative effect of the binding of this compound in the Phe43 cavity in gp120, which is in the proximity of the V3 loop, may have altered the conformation of gp120 to be no longer suitable for CCR5 binding. Consequently, **1** behaves as a CD4 antagonist. Another exciting aspect is that isomer *R*/*S*-(**1**) (NBD-11021A2) inhibits RT, but at a much higher concentration (IC_50_ of 43.4 μM) (Table 1) [37].

In 2016, Curreli and coworkers [38] synthesized twenty-five new analogues of **1**. The X-ray structure of **1** with gp120 reported earlier confirmed that the piperidine ring benefited from enabling its nitrogen atom to hydrogen bond with Asp368 of gp120 [37]. All molecules were tested against HIV-1 in a single cycle and a multicycle infectivity assay. Among these, a substantial improvement in antiviral activity was observed in compounds in which the entire piperidine ring was replaced with a simple primary amine (CH_2_NH_2_), such as compounds NBD-14009 (**3**) and NBD-14010 (**4**) (Figure 5). By replacing the piperidine ring with a methylamine group, the authors reduced solvent exposure of hydrophobic regions and retained the amine for a hydrogen bonding with Asp368 of gp120. They also focused on enhancing surface complementarity between the compound and the cavity [44,45] by replacing the phenyl ring of **1**, which accesses the Phe43 cavity, with a meta-fluoro substituted phenyl ring, as seen in compound **4** (Figure 5) [38]. 

Compound **3** was equipotent to **1** in the single-cycle screening, but less active in the multicycle assay (Figure 5). By comparison, compound **4** proved to be more than two-fold more potent than the prototype in the single-cycle screening (Figure 5), and exhibited better potency than **1** in the multicycle testing (Figure 5). The crystal structure of **4** in complex with HIV-1 gp120 at 2.1 Å resolution confirmed that the meta-fluoro substituted phenyl ring improved surface complementarity with the Phe43 cavity [39]. Both analogs **3** and **4** are less cytotoxic than prototype **1** (Table 1). 

In a continuing effort to discover other analogues of compounds **1** and **2**, the Curreli group [39] designed sixty novel analogues with modifications to regions I and III of compound **1** (Figure 6). Among these, NBD-14088 (**5**) and NBD-14107 (**6**) (Figure 6) displayed excellent activities against HIV-1 (Figure 6). Improvements in the antiviral activity and selectivity index (SI; CC_50_/IC_50_) values by ~8- and 6-fold in the first one-cycle assay were observed (Table 1). The ability of **5** and **6** to inhibit RT was also verified, and both compounds were shown to be more potent than **1**, showing IC_50_ values of 7.2 μM (**5**) and 8.4 μM (**6**) (Table 1) [39]. 

Based on lead compound **6**, Curreli et al. [40] designed a new series of twenty-nine compounds. Through an approach that can enhance interactions with gp120, a change in the CH_2_OH group from C-5 to C-4 in the thiazole moiety was proposed. This change of position was based on observations of the crystal structure of HIV-1 gp120 in complex with **4** that showed important information about the interaction profile [39,40].

A notable improvement in antiviral activity was achieved with NBD-14189 (**7**) (Figure 6) that showed an IC_50_ of 89 nM in a single-cycle assay (standard **6**, 0.64 μM) (Table 1). However, the authors highlighted the need for further improvements concerning its ADMET (absorption, distribution, metabolism, excretion, and toxicity) properties. 

Further optimization of the ADMET properties of lead compound **7** was approached by replacing the phenyl ring in the region I with a pyridine ring (Figure 6) [41], according to the benefits mentioned in the literature [42,43,46]. The best result was observed for NBD-14270 (**8**) (Figure 6). Compared with analog **7**, this new compound showed an increased IC_50_ by approximately two-fold and an improved CC_50_ by approximately five-fold, leading to a SI of 683 (Table 1). 

Exploring region I by introducing bulky molecular groups such as 1,3-benzodioxolyl or its bioisostere 2,1,3-benzothiadiazole, the same group aimed to discover novel entry inhibitors of the NBD class. As a result, this change is well tolerated but does not improve antiviral activity (Table 1) [47]. The best compounds with bulky group substitutions, NBD-14110 (**9**), NBD-14123 (**10**), and NBD-14159 (**11**) (Figure 7), had their anti-RT activity evaluated and moderate inhibition was observed (Table 1).

Compound **12** (Figure 8) was proposed based on guanidine-containing inhibitors as entry antagonists [48]. In the design of this analog, AWS-I-169 (**13**) [45] (Figure 8) and NBD-derivative **4** were used as prototypes. Prototype **13** interacts with the Asp368 residue of gp120, which plays a critical role in binding to Arg59 of the CD4 receptor. 

The results of the biological evaluation of derivative **12** showed improved antiviral activity compared to molecule **13** (compound **13**, IC_50_ 21.3 μM; compound **12**, IC _50_ 10.8 μM) in a single-cycle infection assay (TZM-bl cells/HIV-1_HXB-2_), but this improved activity was not observed when compared with standard NBD analog **4** (IC_50_ 0.59 μM) [49] (Table 1). However, guanidine analog **12** showed an excellent cytotoxicity profile (CC_50_ of 145.9 μM) compared with control molecule **4** (CC_50_ of 40.5 μM) (Table 1) [48]. 

As previously described, gp41 is also responsible for the fusion process between HIV and cells [50]. This glycoprotein is considered one of the most attractive targets for developing HIV entry or fusion inhibitors because of its conserved amino acid sequence [51,52]. Based on this fact, Qiu et al. [53] used NB-64 (**14**) as a prototype in the design of compound **15** as a potential gp41 inhibitor (Figure 9) [54]. The data from the anti-HIV-1 assays of this derivative showed good inhibitory activity in one of the primary strains and an enfuvirtide-resistant strain (primary, IC_50_ 2.078 μM; enfuvirtide-resistant, IC_50_ 5.261 μM), although **15** was less active than the mentioned reference drug (primary, IC_50_ 0.062 μM; enfuvirtide-resistant, IC_50_ > 2 μM) (Figure 9).

## 3. Pyrrole-Based Anti-HIV-1 Inhibitors (Dual Inhibitors, RT Inhibitors and Replication Inhibitors)

The therapeutic approach to treat AIDS with a drug cocktail is often compromised by low patient compliance and the risk of drug–drug interactions [55]. Therefore, the development of multitargeted molecules would be an alternative to simultaneously modulate different targets, eliminating the occurrence of drug–drug interactions and facilitating good adherence to therapy [56]. Molecules that target two enzymes are called dual inhibitors [57], and the development of these types of inhibitors is extremely interesting for AIDS management.

In this respect, some works have described the development of dual inhibitors targeting essential enzymes in the replication cycle of HIV-1, such as IN and RT [57,58]. IN is responsible for integrating the DNA of HIV-1 into the host genome. This process depends on the assistance from two Mg^2+^ ions that act as cofactors in the catalytic core domain (CCD) of IN. The catalytic site of the ribonuclease H domain (RNase H) of HIV-1 RT, which hydrolyzes the RNA strand of the RNA-DNA hybrid, resembles the topology found in the CCD of IN [59]. The hydrolytic process involves water molecules that act as nucleophiles and magnesium ions coordinating with the highly conserved residues of RNase H (Asp443, Asp498, Asp549, and Glu478) [60]. Due to this similarity, these two enzymes are a potential duet for the discovery of HIV-1 dual inhibitors.

Pyrrolyl scaffold is a good starting point to obtain molecules with promising anti-IN or anti-IN/RNase H pharmacological profiles once it has a synthetic versatility [61]. In this context, two pyrrolyl derivatives, **16** and **17**, were developed [9,57]. Compounds **16** and **17** showed dual activity with micromolar IC_50_ values against HIV-1 IN and RT RNase H. Notably, pyrrole **17** was a promising dual inhibitor, although it did not exhibit good potency in HIV-infected cells (**16**, EC_50_ 2 μM; **17**, EC_50_ 20 μM) and both of them showed a CC_50_ > 50 μM (Figure 10). Comparing the selectivity pattern of the two compounds, **16** provides excellent and greater cell protection (SI > 25 μM) than **17** (SI > 2.5 μM).

Intending to overcome the problems reported for diketo acid (DKA) [62] and improve selectivity for RNase H of HIV-1, Messore et al. [58] developed a library of compounds that mimicked the DKA moiety presented in dual inhibitor **18** (IC_50_ 10 μM against HIV-1 RNase H) (Figure 11) through replacement with a pyrazole ring. In this series, derivative **19** (Figure 11) was the most potent compound, showing an IC_50_ of 0.27 μM against HIV-1 RNase H (Figure 11). Additionally, the goal of obtaining better selectivity for the RT domain versus IN was achieved, and the derivatives showed good serum stability compared to their corresponding DKA derivatives.

RT plays an important role in the HIV-1 replication cycle. RNA-dependent DNA polymerase activity (RDDP), DNA-dependent DNA polymerase activity (DDDP), and RNase H function make RT a multifunctional enzyme. However, there are not any molecules targeting RNase H in RT approved by the FDA [63]. The DNA polymerase (pol) domain, in addition to the RNase H domain, is located at the catalytic p66 subunit of HIV-1 RT. RNase H domain is essential for viral genome replication by carrying out the hydrolysis of the viral RNA strand in the RNA/DNA hybrid [60], and studies show that its inhibition completely blocks viral replication [60,64,65,66].

RNase H inhibitors are a class of compounds that present critical issues that hamper drug development. First, the nucleic acid substrate (RNA/DNA hybrid) binds to p66 and interacts with both the pol and the RNase H active sites. Due to the reciprocal action between the two functions of RT, it is a challenge to achieve the selective RNase H inhibition over pol [67]. As previously mentioned, RT requires a coordinate binding of the RNA/DNA substrate to both pol and RNase H domains; because the majority of the binding interactions are in the pol domain, there is a low binding dependence of the substrate on the RNase H domain [67,68].

In addition, RT RNase H belongs to the retroviral integrase superfamily, and the similarities complicate efforts toward selective inhibition of RNase H over IN. Despite the problems reported for effective inhibition of the RNase H domain of the HIV-1 RT, papers with this goal have been published [69,70,71,72]. 

In 2017 [73], a library of 1*H*-pyrrole-3-carbothioamide and 1*H*-pyrazole-4-carbothioamide derivatives were synthesized with the aim of obtaining molecules that are able to promote the simultaneous inhibition of all points of RT. The compounds were screened against HIV-1 (IIIB strains/MT-2 cells), and their RT-associated RNase H and RDDP inhibitory activities were measured using the methodology described by Corona et al. [74] (using prototype foamy viruses) and Meleddu et al. [75] (using an Invitrogen EnzCheck Reverse Transcriptase Assay Kit, Eugene, Or, USA), respectively.

Among the pyrrole derivatives, molecule **20** showed the best RT-associated RNase H inhibitory activity, with an IC_50_ of 6.1 ± 1.6 μM. However, its EC_50_ value for HIV-1 inhibition activity in the cell-based assay was >30 μM (Figure 12), demonstrating its lack of ability to inhibit HIV-1 replication. In addition, all tested pyrrole derivatives were cytotoxic (Figure 12). Due to its inability to maintain cell viability, the RT-associated RDDP inhibitory activity of these pyrrole derivative was not verified. Only the most active compound, which belonged to the 1*H*-pyrazole-4-carbothioamide derivative series, was evaluated by the authors for its inhibitory potential of RT-associated RDDP [73]. 

In search of new RT inhibitors of HIV-1, in 2019, certain copper (Cu) complexes that target this viral enzyme were reported (Figure 13) [76]. The literature described that metal chelating agents represent a strategy for the development of potential enzyme inhibitors [77,78,79]. Therefore, three pyrrole-copper complexes were synthesized (**21**–**23**) and found to be effective inhibitors of HIV-1 RT (Figure 13). Complex **21** had a percentage of HIV-1 RT inhibition (70.18%) that was near that of the standard nevirapine (100%) at a concentration of 221.69 μM (Table 2). At lower concentrations, complex **22** showed better results, with 10.82% and 46.54% inhibition activity at 21.68 and 108.43 μM, respectively (Table 2). Regarding cytotoxicity, only complex **21** considerably decreased cell viability at all tested concentrations (Table 2).

Stanton and coworkers [80] identified ten novel antiretroviral compounds built with a 7-azaindole (pyrrolo[2,3-b]pyridine) core (Figure 14) by testing a library composed of 585 molecules. The antiretroviral activity of these compounds was determined in physiologically relevant primary human peripheral blood mononuclear (PBM) cells infected with HIV-1 (strain LAV-1) [81].

The data from Figure 14 reveal that all compounds showed submicromolar anti-HIV-1 potency. To determine the action mechanism of the most active compound **24**, its ability to inhibit polymerase RT activity was measured. The IC_50_ detected for 24 was 0.73 ± 0.32 μM (Figure 14), and the authors concluded that this molecule acts as a non-nucleoside reverse transcriptase inhibitor (NNRTI); after increasing the nucleotide concentration, the observed IC_50_ did not decrease. In contrast, the potency verified for **24** was enhanced by adding natural nucleotides. Evaluation against mutant strains (K103N, Y181C, and E138K) resulted in some degree of activity; however, compound **24** was not able to act against the V108I mutant strain [80].

After evaluation by the free energy perturbation (FEP) method, modification of the lower phenyl ring of **24** was proposed in order to increase binding affinity and hence potency against RT. Therefore, compound **34** was synthesized and demonstrated improved potency against the HIV-1 enzyme, which was two-fold greater than that of **24** (IC_50_ 0.36 ± 0.01 μM) (Figure 15). Although more potent than **24** against wild-type RT, the new derivative **34** demonstrated some activity against the Y181C RT mutant and was inactive against RT mutant forms K103N, V108I, and E138K [80].

In an attempt to obtain active compounds in cells infected by HIV-1, Yonn et al. [82] synthesized new scaffolds of pyrrolo-pyridine derivatives (Figure 16). The *N*-substituted pyrroles **35**–**37** (Figure 16) showed moderate activity against HIV-1 (EC_50_ values ranging from 5.02 to 5.07 μM) when compared with the control drugs (AZT, EC_50_ 0.0063 μM; raltegravir, EC_50_ 0.0063 μM) (Table 3).

Recently, cyclic amides, known as lactams, coupled with pyrrole have proven to be a promising group in the search for an anti-HIV drug candidate, as demonstrated by compound **38** (Figure 16) [83]. Within a set of six compounds, **38** was the only compound active at a noncytotoxic concentration, showing an EC_50_ of 2.74 ± 1.08 μM (CC_50_ 18.93 ± 4.0 μM) against HIV-1; however, **38** was inactive against replication in the HIV-2 cell line (Table 3). The docking analysis carried out with crystal structures of RT of HIV-1 and HIV-2 suggested that compound **38** was a potent HIV-1 NNRTI agent [83].

As shown, pyrrole-based anti-HIV compounds are promising therapeutic agents for the treatment of AIDS. In this work, we attempted to review the literature involving the development of pyrrole analogs as anti-HIV agents in the time period between 2015 and 2020.

## 4. Concluding Remarks and Future Directions

AIDS therapy faces a series of drawbacks, mainly because its treatment is based on a cocktail that is difficult for patients to adhere to. Moreover, other significant problems exist, such as long-term toxicity, the emergence of drug-resistant viral strains, and drug–drug interactions. In this context, there is a continuous need to identify new anti-HIV agents based on several chemical groups, including nitrogenated heterocycles such as pyrroles. Among the compounds with potent antiretroviral activity, designs based on the pyrrole scaffold is significant. This was confirmed with fostemsavir, which is the first antiretroviral drug used against HIV-1, and exhibits a pyrrole ring in its structure, showing the importance of exploring this nucleus.

It is possible to predict that pyrrole-based antiretrovirals can be further exploited because they may provide compounds with high activity in the micromolar or nanomolar range. In addition, it was observed that targeting the Asp368 residue of gp120 may possibly lead to the design of molecules that are attractive and innovative pyrrole drugs as entry inhibitors. Another promising approach using this ring is in the field of multitargeted molecules focused on the enzymes IN and RT, because these enzymes share similar topological features in their catalytic sites. As a result, certain compounds containing pyrrole rings with interesting in vitro and ADMET profiles are potential candidates for new antiretroviral drugs.

In summary, this review analyzed the progress between 2015 and 2020 regarding the antiretroviral activity of molecules containing a pyrrole ring, which is considered a privileged scaffold for drug discovery.

## Figures and Tables

**Figure 1 pharmaceuticals-14-00893-f001:**
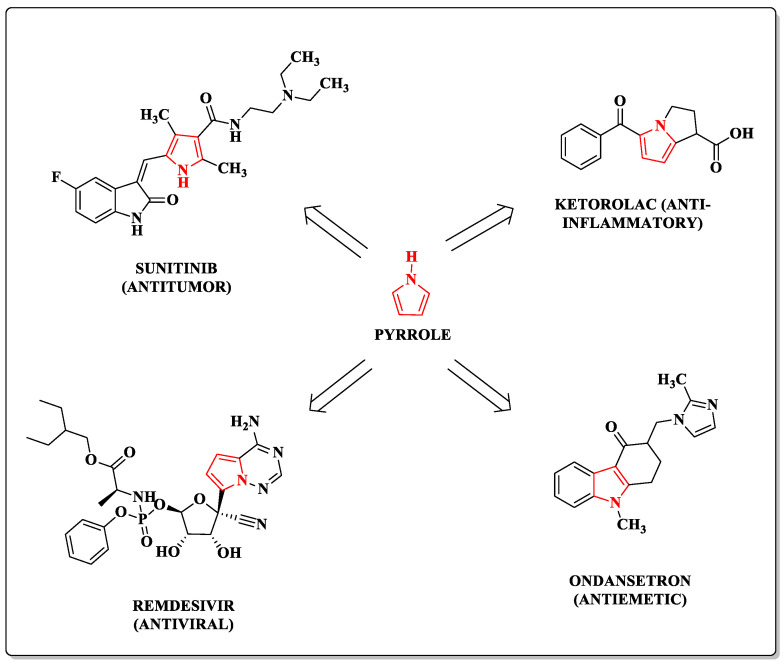
Examples of drugs containing pyrrole moieties approved by the FDA.

**Figure 2 pharmaceuticals-14-00893-f002:**
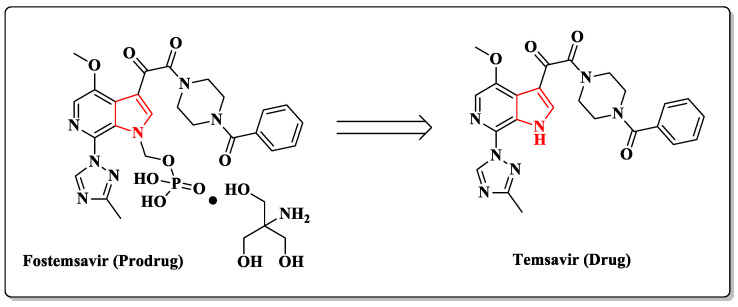
Chemical structures of temsavir and its prodrug fostemsavir, highlighting the pyrrole ring in red.

**Figure 3 pharmaceuticals-14-00893-f003:**
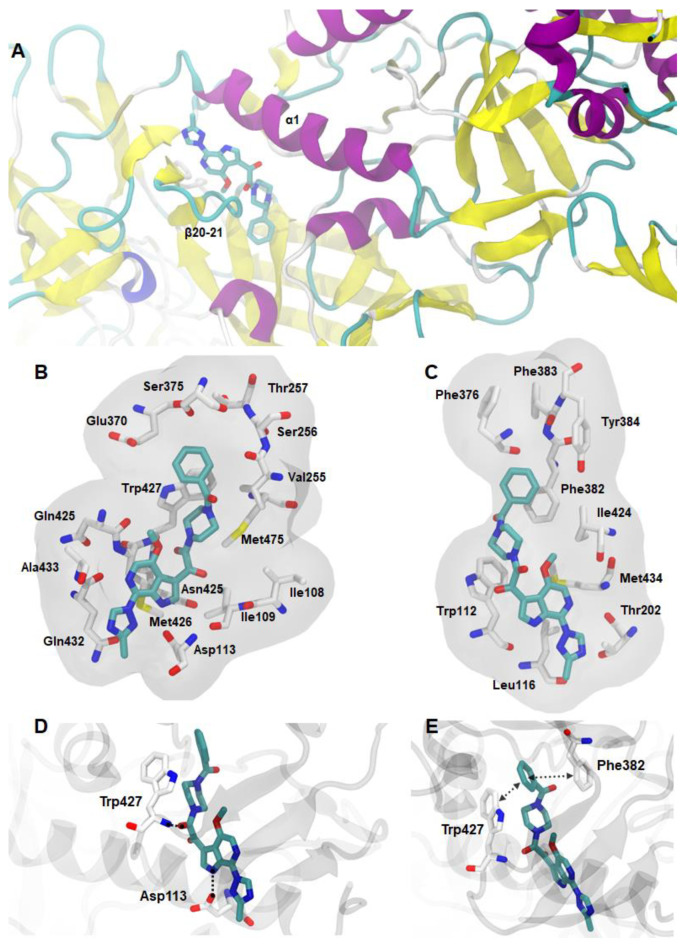
(**A**) Representations of the temsavir binding site into gp120, showing the β20-21 loop (β20-21) and α1-helix (α1), and the interactions between temsavir and gp120: (**B**,**C**) steric, (**D**) hydrogen bond, and (**E**) aromatic-stacking interactions. The hydrogen bonds and aromatic-stacking interactions are represented by black interrupted lines. The gp120 protein is in cartoon representation, colored by secondary structure (β-sheet in yellow, α-helix in purple, turn in cyan, and random structure in white). Temsavir and gp120 residue structures are shown in the stick models and colored by atom (the nitrogen atoms are shown in blue, the oxygen atoms in red, the sulfur atoms in yellow, and the carbon chain in white or cyan).

**Figure 4 pharmaceuticals-14-00893-f004:**
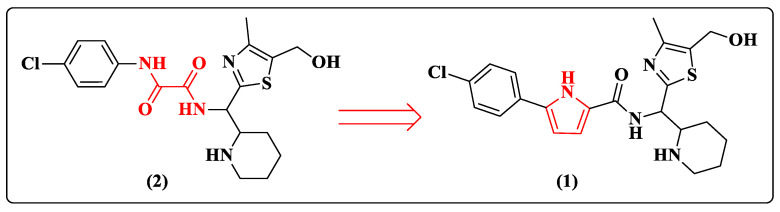
Chemical structure of compounds **1** and **2** [37].

**Figure 5 pharmaceuticals-14-00893-f005:**
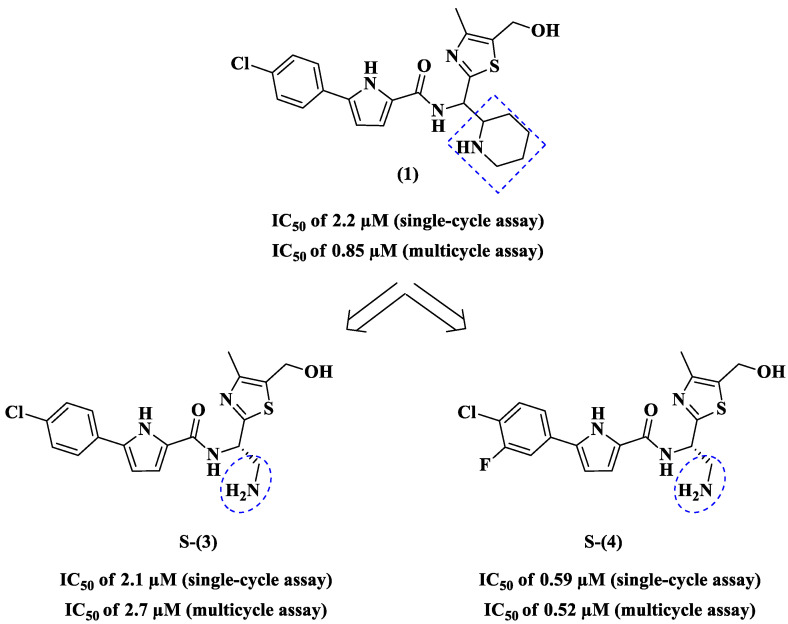
Structures of **1**, **3**, and **4** and their anti-HIV-1 profiles [37,38].

**Figure 6 pharmaceuticals-14-00893-f006:**
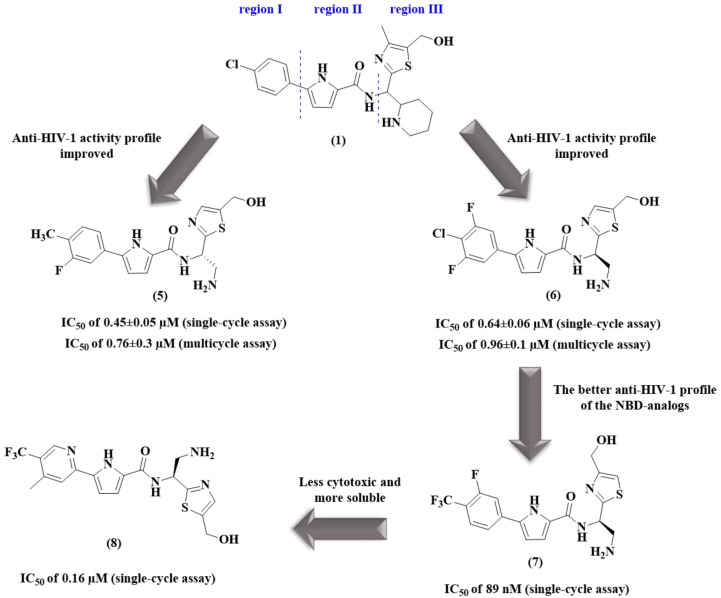
Structural modifications of compound **1** that led to analogs **5**–**8** and their anti-HIV-1 profiles [39,40,41].

**Figure 7 pharmaceuticals-14-00893-f007:**
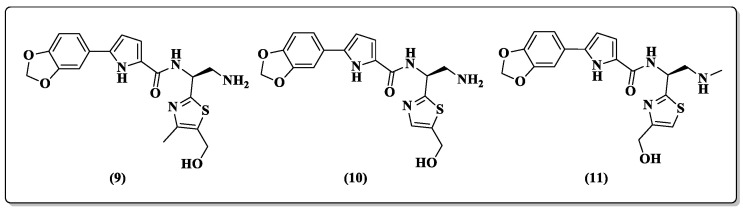
Chemical structures of the three best compounds synthesized by Curreli et al. [47]: **9**, **10**, and **11**.

**Figure 8 pharmaceuticals-14-00893-f008:**
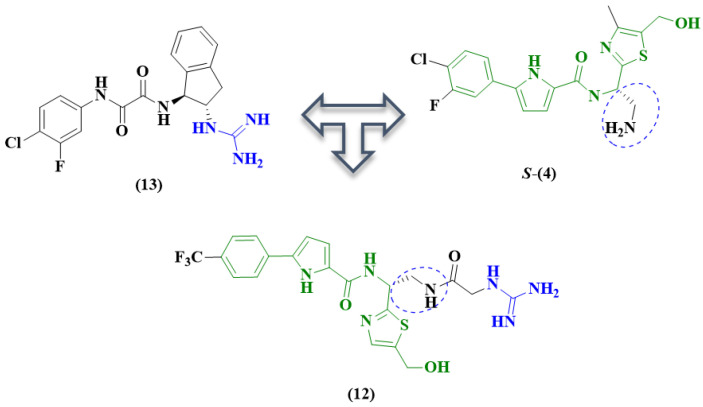
Design of compound **12** [45,48].

**Figure 9 pharmaceuticals-14-00893-f009:**
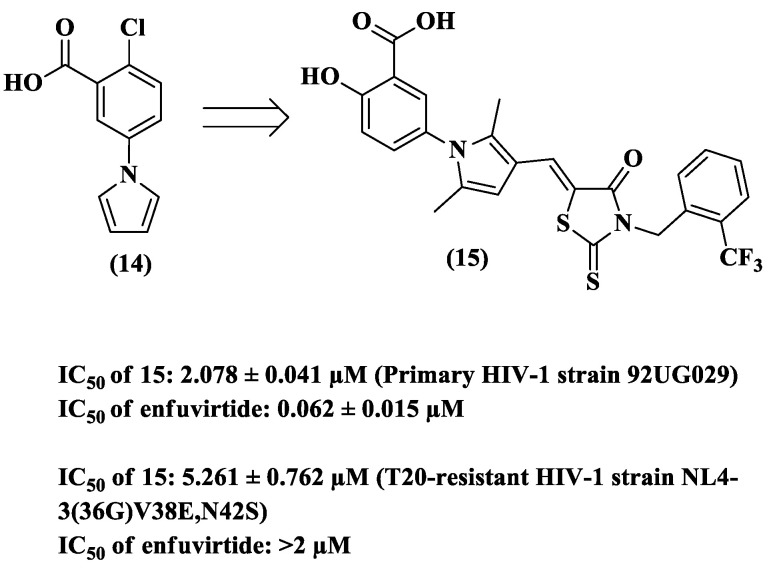
Chemical structures of **14** and **15** and the anti-HIV-1 profile of derivative **15** [53].

**Figure 10 pharmaceuticals-14-00893-f010:**
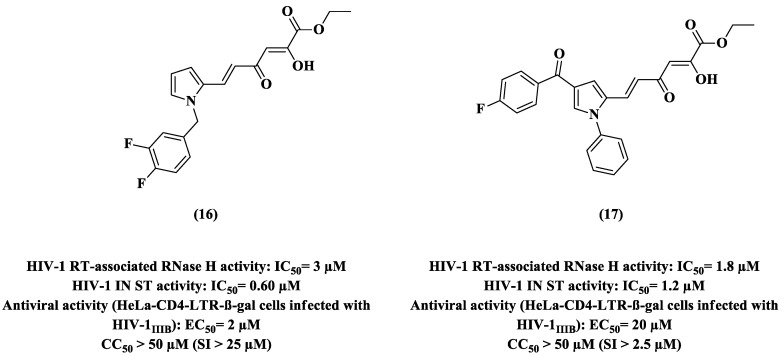
Chemical structures of **16** and **17** and their anti-HIV-1 profiles [57].

**Figure 11 pharmaceuticals-14-00893-f011:**
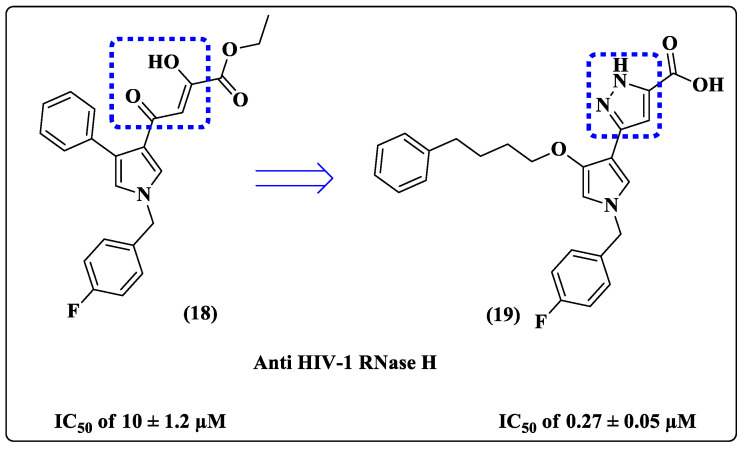
Chemical structures of **18** and **19** and their anti-HIV-1 RNase H activities [58].

**Figure 12 pharmaceuticals-14-00893-f012:**
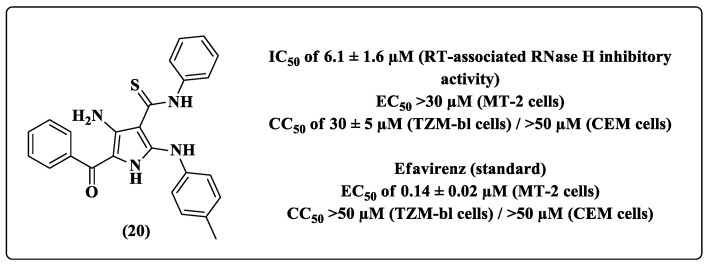
Chemical structure and anti-HIV-1 profile of compound **20** [73].

**Figure 13 pharmaceuticals-14-00893-f013:**
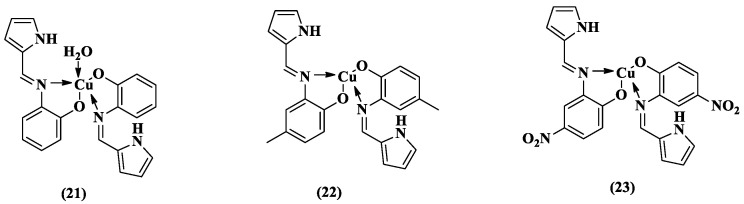
Chemical structures of the Cu(II) complexes of **21**–**23** [76].

**Figure 14 pharmaceuticals-14-00893-f014:**
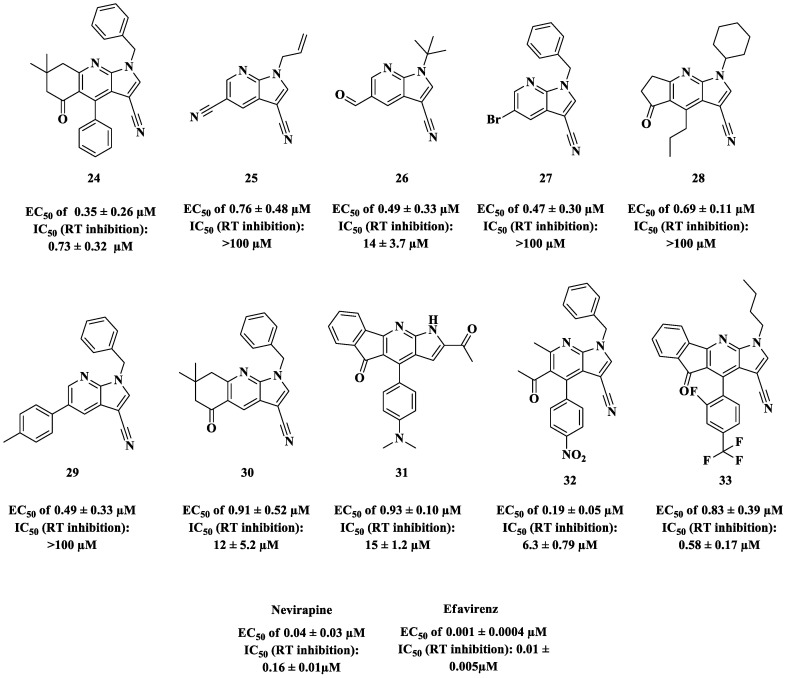
The ten most potent 7-azaindoles from a library of molecules investigated by Stanton and coworkers [80].

**Figure 15 pharmaceuticals-14-00893-f015:**
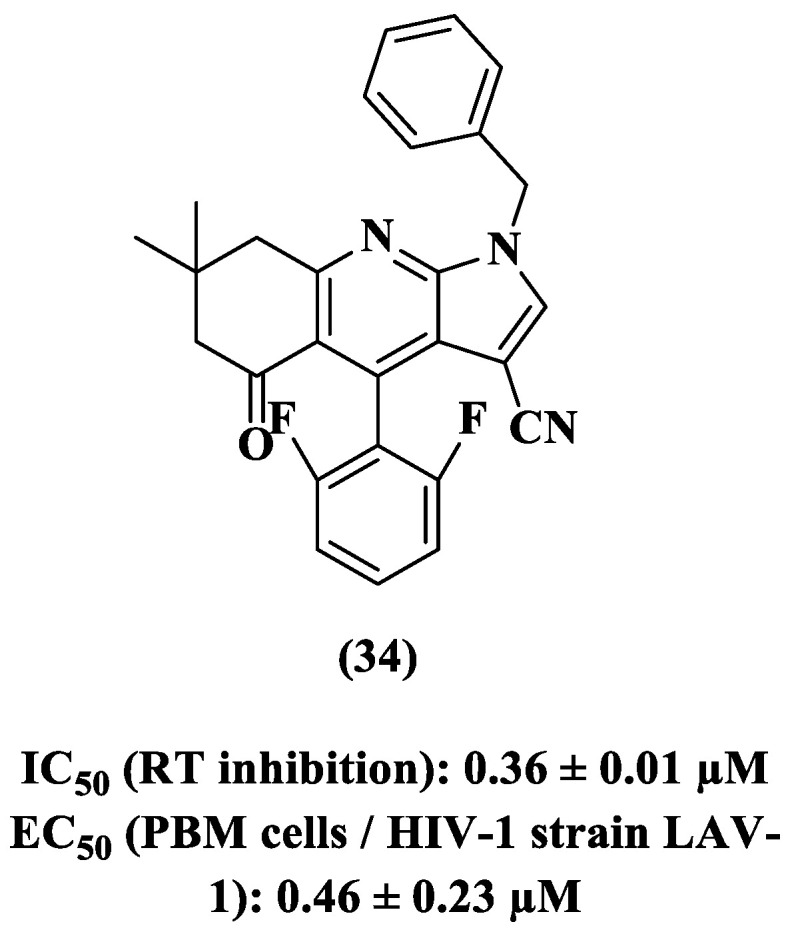
Chemical structure and anti-HIV-1 profile of compound **34** [80].

**Figure 16 pharmaceuticals-14-00893-f016:**
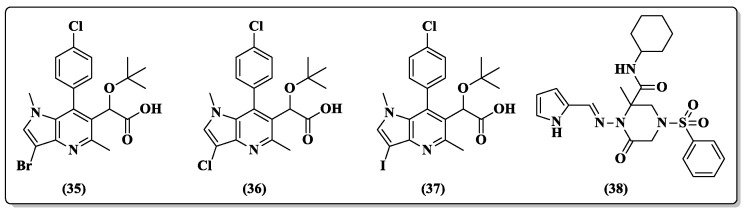
Chemical structure of compounds **35**–**38** [82,83].

**Table 1 pharmaceuticals-14-00893-t001:** Anti-HIV-1 activity and cytotoxicity of CD4 antagonists in single-cycle (TZM-bl cells) and multicycle (MT-2 cells) assays, selectivity index (SI), and inhibition of HIV-1 RT [37,39,40,41,47,48].

NBD- Analog	TZM-bl Cells (Mean ± SD) ^a^		MT-2 Cells (Mean ± SD) ^a^		SI ^b^	Inhibition of HIV-1 RT
	IC_50_ (μM)	CC_50_ (μM)	IC_50_ (μM)	CC_50_ (μM)		IC_50_ (μM)
**1**	2.2 ± 0.2 ^d^/2.5 ± 0.2 ^f^	23.6 ± 1.1 ^d^/~28 ^f^	0.85 ± 0.06 ^d^/1.5 ± 0.06 ^f^	24.8 ± 2.3 ^d^/~28 ^f^	10.7 ^d^/~11.2 ^f^	47 ^d^/43.4 ^f^
**2 ^f^**	4.7 ± 1.1	23.7 ± 1.1	4.7 ± 0.6	>108	5.04	-
***S*-(3)**	2.1 ± 0.2	33.6 ± 0.8	2.7 ± 0.33	25.8 ± 8.8	-	-
***S*-(4)**	0.59 ± 0.06	40.5 ± 1.3	0.52 ± 0.05	30.5 ± 0.6	-	-
**5**	0.45 ± 0.05	38.8 ± 0.8	0.76 ± 0.3	38 ± 1	86.2	7.2
**6**	0.64 ± 0.06	39.5 ± 2.3	0.96 ± 0.1	37 ± 1.5	61.7	8.4
**7**	0.089 ± 0.001	21.9 ± 0.5	0.18 ± 0.001	22.1 ± 1	246	NT ^c^
**8**	0.16 ± 0.004	109.3 ± 2	NT ^c^	NT ^c^	683.1	NT ^c^
**9**	2.3 ± 0.1	145.6 ± 7.6	3.8 ± 1.4	193 ± 6.3	63.3	30.6 ± 2.7
**10**	4.3 ± 0.8	142.3 ± 2.6	4.5 ± 0.9	191.5 ± 2.6	33.1	39.5 ± 2.8
**11**	2.7 ± 0.3	95.8 ± 1.4	2.98 ± 0.1	180 ± 2	35.5	28.1 ± 7.9
**12 ^g^** [48]	10.8 ± 0.6 μM	145.9 ± 8.9 μM	NT ^c^	NT ^c^	-	NT ^c^
**13 ^g^**	21.3 ± 5.0 μM	-	NT ^c^	NT ^c^	-	NT ^c^
Temsavir	<1 nM ^e^	>100 ^e^	NT ^c^	NT ^c^	-	NT ^c^

^a^ The reported IC_50_ and CC_50_ values represent the means ± standard deviation (SD) of three replicates; ^b^ Selectivity index (SI) from the single-cycle assay: CC_50_/IC_50;_
^c^ NT: Not Tested; ^d^ [39]; ^e^ [41]; ^f^ [37]; ^g^ [48].

**Table 2 pharmaceuticals-14-00893-t002:** Anti-HIV-1 RT activity and cell viability of Cu(II) complexes (**21**–**23**) [76].

Compound	% of Inhibition (Tested Concentration)			% of Cell Viability			
				10 μM	50 μM	100 μM	150 μM
**21**	2.11 (22.16 μM)	46.54 (110.84 μM)	70.18 (221.69 μM)	38	32	25	20
**22**	10.82 (21.68 μM)	46.54 (108.43 μM)	67.54 (216.87 μM)	100	100	100	100
**23**	6.34 (19.11 μM)	35.17 (95.59 μM)	48.44 (191.18 μM)	100	100	100	100

**Table 3 pharmaceuticals-14-00893-t003:** Antiviral evaluation of compounds **35**–**38** in MT-4 cells [82,83].

Compound	Antiviral Activity (EC_50_) (μM)			CC_50_ (μM) ^b^	
	HTLV-III_B_ ^a^	HIV-1, IIIB ^b^	HIV-2, ROD ^b^	HIV-1, IIIB ^b^	HIV-2, ROD ^b^
**35**	5.02	NT ^c^	NT ^c^	NT ^c^	NT ^c^
**36**	5.03	NT ^c^	NT ^c^	NT ^c^	NT ^c^
**37**	5.07	NT ^c^	NT ^c^	NT ^c^	NT ^c^
**38**	NT ^c^	2.74 ± 1.08	>17.13	18.93 ± 4.0	>18.95
AZT	0.0063	NT ^c^	NT ^c^	NT ^c^	NT ^c^
Raltegravir	0.0063	NT ^c^	NT ^c^	NT ^c^	NT ^c^
Efavirenz	NT ^c^	0.00441	NT ^c^	>6.327	NT ^c^
Nevirapine	NT ^c^	0.198	NT ^c^	>14.02	NT ^c^

^a^ [82]; ^b^ [83] ^c^ NT: Not Tested.

## Data Availability

Data sharing not applicable.

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
