# Peer review of "Pyrroles as Privileged Scaffolds in the Search for New Potential HIV Inhibitors"

_pharmaceuticals, 2021, doi:10.3390/ph14090893_

Round 1

Reviewer 1 Report

The manuscript is intended to describe pyrrole-based compounds with anti-HIV activity between 2015 and 2020. However, the manuscript is poorly executed as below. The figures are not properly organized and made confusing by different data, colors, frames, arrows, tables et. al.. The tables or are messed by compound structures or lose compound numbers (such as Table1 and Table 5). The manuscript format is confused including disordered structure sizes, data format, reference locations. Totally, the manuscript is not been well organized and very confusing. 

Author Response

The manuscript is intended to describe pyrrole-based compounds with anti-HIV activity between 2015 and 2020. However, the manuscript is poorly executed as below. The figures are not properly organized and made confusing by different data, colors, frames, arrows, tables et. al. The tables or are messed by compound structures or lose compound numbers (such as Table1 and Table 5). The manuscript format is confused including disordered structure sizes, data format, reference locations. Totally, the manuscript is not been well organized and very confusing. 

Response: Dear Sir, we appreciate the comments and the well-done revision. We want to report that changes were made in all figures trying to make them more straightforward for the reader. Besides, the font, size of the figures, and the chemical structures have also been revised and modified. We also standardize the format of the tables with no chemical structures. When possible, we consolidated data from different tables into a single table, such as data from table 2 were grouped in table 1, as well as data in table 5 were grouped with data in table 4 from the original manuscript. We hope that those changes made the manuscript more organized and easier for the comprehension of the readers.

Reviewer 2 Report

This review article from Bianco et al. summarizes recent efforts to develop single molecule inhibitors of HIV-1 replication based on pyrrole chemistry. The article summarizes how pyrrole “scaffolds” have been used to design antiretroviral drugs targeting HIV-1 entry and reverse transcription. From a virologist’s perspective, the review is interesting and provides a thoughtful perspective on how pyrroles can serve as a modular core for building target specificity and improving pharmacokinetics. On the other hand, the review would be much improved with better organization with suggestions outlined below.

Major comments:

1. First, most of the figures feel extraneous, often repeating information that is already in  tables. Figure details could be consolidated, and would ideally be only used to illustrate, concisely, major themes for a reader. The intended format for the tables in general seems fine, but as presented some data are missing and it would be better to consider consolidating the data into fewer tables based on distinct themes, so that the comparisons are easier to navigate.

2. Second, the text is not very well-balanced. The bulk of the story is a relatively detailed description of pyrrole derivatives inspired by the entry blocker temsavir (pages 4-10); flanked by a short, relatively sparse introduction to the relevant medicinal chemistry and a concise summary of efforts to use pyrroles to target Reverse Transcriptase and Integrase. The section on entry inhibitors could be shortened, and the other two sections expanded, in particular with the introduction saying much more about the pros and cons of pyrrole chemistry in the context of screens and rational design, and in particular a stronger case for why the authors feel pyrroles should be considered over other scaffolding cores in the context of antiviral drug development. In the end, most of the drugs perform in the µM range in cell culture, which doesn’t seem to bode particularly well for anticipated in vivo efficacy. Moreover, it seems that a lot of structural work is still needed to even begin understand substrate binding specificity- a point that should be emphasized and discussed further. It also troubles me that if some of the drugs are binding both gp120 and RT, that are not similar in structure or function, then for many of these drugs there are likely to be major off-target effects.

Minor comments:

1. Introduction:

  1. The statement “38 million people had been infected by this retrovirus.” is not accurate (probably twice that over 4 decades). Are the authors referring to those currently infected?
  2. Clause “..the emergence of drug-resistant viral strains” at end of paragraph 2- redundant with prior statement.

2. Section 2: 

    1. Paragraph 2- Probably want to point out that is a pyrrole drug earlier in the paragraph. And describe how the pyrrole chemistry is important to the prodrug (and how it works).
    2. Paragraph 3- “steric interactions”-aren’t all molecular interactions steric?
    3. Paragraph 4- no real need to reiterate the need for new drugs here.
    4. Paragraph 6- text implies that (1) blocks entry by binding to both gp120 and CD4, and also blocks RT. How is this possible?

3. Section 3:

    1. Text makes it sounds as if drug has RDDP activity, intended to say that inhibits RDDP, not having RDDP activity?
    2. For RNAse H, RDDP, and DDDP inhibition, how were the specific measurements made? Please describe.

4. Fig1- I did not understand the point of the partitioned circle and the arrow in this figure. What are the authors trying to communicate specifically?

5. Fig 3- For A and B the context of the block to infection is not really explained- and the structure info is cool but a bit arbitrary if you can't see all of Env to understand how the drug is inhibiting Env-CD4 binding.   

6. Fig4- Text says that (2) actually enhances infection but the figure shows an IC50? Please resolve.   

7. Fig5- Text says that (3) and (4) are less cytotoxic than (2) but CC50s here and in table don’t seem to be better?

8. Fig7- "P-gp" substrate and CYP1 details are not explained for the reader.  

9. Fig9- Please define T-20 for the reader.

10. There are some some issues with some of the Tables in the submitted pdf- don’t show the compounds. Also, should consider combining Tables 1 and 2, if not others.

11. Text frequently says “prevents cell-to-cell transmission” but what specific assay are the authors referring to? If referring to the multi cycle assay or cell-cell fusion assay- neither is really a cell-to-cell transmission assay.

Author Response

Point 1: First, most of the figures feel extraneous, often repeating information that is already in tables. Figure details could be consolidated, and would ideally be only used to illustrate, concisely, major themes for a reader. The intended format for the tables in general seems fine, but as presented some data are missing and it would be better to consider consolidating the data into fewer tables based on distinct themes, so that the comparisons are easier to navigate.

Response: Dear Sir, we appreciate the comments and the well-done revision. We want to report that changes were made in all figures trying to make them more straightforward for the reader. Besides, the font, size of the figures, and the chemical structures have also been revised and modified. We also standardize the format of the tables with no chemical structures. When possible, we consolidated data from different tables into a single table, such as data from table 2 were grouped in table 1, as well as data in table 5 were grouped with data in table 4 from the original manuscript. We hope that those changes made the manuscript more organized and easier for the comprehension of the readers.

Point 2: Second, the text is not very well-balanced. The bulk of the story is a relatively detailed description of pyrrole derivatives inspired by the entry blocker temsavir (pages 4-10); flanked by a short, relatively sparse introduction to the relevant medicinal chemistry and a concise summary of efforts to use pyrroles to target Reverse Transcriptase and Integrase. The section on entry inhibitors could be shortened, and the other two sections expanded, in particular with the introduction saying much more about the pros and cons of pyrrole chemistry in the context of screens and rational design, and in particular a stronger case for why the authors feel pyrroles should be considered over other scaffolding cores in the context of antiviral drug development. In the end, most of the drugs perform in the µM range in cell culture, which doesn’t seem to bode particularly well for anticipated in vivo efficacy. Moreover, it seems that a lot of structural work is still needed to even begin understand substrate binding specificity- a point that should be emphasized and discussed further. It also troubles me that if some of the drugs are binding both gp120 and RT, that are not similar in structure or function, then for many of these drugs there are likely to be major off-target effects.

Response: Dear Sir, we would like to inform the reviewer that we expanded the section 3 introduction, as shown in paragraphs 6, 7, and 8. Besides, paragraph 3 (also included) highlights why the authors feel pyrroles should considered in antiviral drug development. We agree with the observation about the anticipated in vivo efficacy, and we corrected this sentence in the conclusion section. If some compounds can bind in more than two targets, off-target effects should expected. However, considering the therapeutic approach to AIDS, perhaps this fact is interesting, enabling a reduction in the number of pills administered in therapy. For this reason, we decided to explain in the manuscript the importance of multi targets in the treatment of multifactorial diseases and based on cocktails.

Point 3: The statement “38 million people had been infected by this retrovirus.” is not accurate (probably twice that over 4 decades). Are the authors referring to those currently infected?

Response: Dear Sir, we are referring to the number of people infected by HIV in 2019. The correction in the text was made.

Point 4: Clause “the emergence of drug-resistant viral strains” at end of paragraph 2- redundant with prior statement.

Response: DONE

Point 5: Paragraph 2- Probably want to point out that is a pyrrole drug earlier in the paragraph. And describe how the pyrrole chemistry is important to the prodrug (and how it works).

Response: Done. Insertion of paragraph 3.

Point 6: Paragraph 3- “steric interactions”- aren’t all molecular interactions steric?

Response: Dear reviewer, in that paragraph, we only describe all the interactions reported by Pancera and coworkers in their published paper (https://www.nature.com/articles/nchembio.2460), including steric, H-bond, and aromatic stacking interactions.

Point 7: Paragraph 4- no real need to reiterate the need for new drugs here.

Response: Done.

Point 8: Paragraph 6- text implies that (1) blocks entry by binding to both gp120 and CD4, and also blocks RT. How is this possible?

Response: Dear Sir, actually (1) blocks entry through binding to the CD4 cell receptor and it implies in the block of the binding of HIV-1 gp120 at this receptor. Compound (1) does not block gp120. That is the main action mechanism, but the authors revealed some degree of activity by RT inhibition.

Point 9: Text makes it sounds as if drug has RDDP activity, intended to say that inhibits RDDP, not having RDDP activity?

Response: Done.

Point 10: For RNAse H, RDDP, and DDDP inhibition, how were the specific measurements made? Please describe.

Response: Done.

Point 11:  Fig1- I did not understand the point of the partitioned circle and the arrow in this figure. What are the authors trying to communicate specifically?

Response: Done. Figure modificated.

Point 12: Fig 3- For A and B the context of the block to infection is not really explained- and the structure info is cool but a bit arbitrary if you can't see all of Env to understand how the drug is inhibiting Env-CD4 binding.   

Response: Dear reviewer, in that figure, we add an image (Fig 3A) showing the binding site of temsavir into the gp120 and a text describing the features of its binding site.

Point 13: Fig4- Text says that (2) actually enhances infection but the figure shows an IC50? Please resolve.   

Response: Done.

Point 14: Fig5- Text says that (3) and (4) are less cytotoxic than (2) but CC50s here and in table don’t seem to be better?

Response: Dear Sir, (3) and (4) are less cytotoxic than (1). As shown in table 1 both molecules exhibit CC50 values higher than those shown by 1, which proves their better cell protection profile.

Point 15:  Fig7- "P-gp" substrate and CYP1 details are not explained for the reader.  

Response: Done.

Point 16: Fig9- Please define T-20 for the reader.

Response: Done.

Point 17: There are some some issues with some of the Tables in the submitted pdf- don’t show the compounds. Also, should consider combining Tables 1 and 2, if not others.

Response: Done.

Point 18: Text frequently says “prevents cell-to-cell transmission” but what specific assay are the authors referring to? If referring to the multi cycle assay or cell-cell fusion assay- neither is really a cell-to-cell transmission assay.

Response: Dear Sir, as explained in caption C of table 1, this specific assay is made through cocultured MAGI-CCR5 cells with Env-/Tat-expressing HL2/3 cells.

Reviewer 3 Report

The  article by Bianco et al. presents a review and drug-like applicability of pyrrole-containing compounds that have potential to become anti-HIV compounds. The authors did a comprehensive analysis of these compounds and suggested that pyrrole ring-containing compounds can target multiple viral proteins including gp120 and RNase H domain of HIV-1 reverse transcriptase. In my opinion, it is an interesting review. However, the experimental data show that the compounds have high IC50 and it is not likely that these compounds will ever get to the clinical trials. The bar for developing anti-HIV compounds is very high. With the recent approval of cabotegravir and rilpivirine combination that acts a long-acting treatment, the focus of the anti-HIV drug development field is on the compounds that can be administered as long-acting compounds. That includes the new capsid inhibitor by GILEAD and Islatavir by MEERCK. Having said that, the door for new drugs is not necessarily closed. With this review presenting new perspectives, makes the field quite innovative and interesting. Thus, the review is worth publishing. However, there are some concerns that need to be addressed.

  1. The major concern is that it is unfair to call a compound pyrrole based when the pyrrole ring is fused to other ring. For example, in RDV (remdesevir), it is fused with another ring. That is the reason in IUPAC nomenclature is named as 4-aminopyrrolo. Additionally, RDV is not necessarily a pyrrole based drug. Also, only one of the three examples in Fig. 1 qualify as being the pyrrole-based drug. The same stands true for two entry inhibitors shown in Figure 2. The authors should be cautious to call these drugs as pyrrole-based.
  2. More details about the binding of the two entry inhibitors should be provided. The authors may want to include the location of V2/V3 loops of gp120 and a famous phenylalanine (F43) of CD4 and BMS-626529 bind.
  3. RNase H inhibitors are a long shot as anti-HIV drugs. This is because it is not clear at what step of viral replication, these compounds can act. Most certainly, these compounds are not good enough to compete with DNA/DNA or DNA/RNA complex of HIV-1 RT. Besides, there is no obvious binding pocket for pyrrole containing compounds to the RNase H domain of HIV-1 RT. Authors should comment on this fact. 
  4. Only compound 4 has moderate IC50. The authors are suggested to comment on the IC50 of these compounds and suggest the modifications that can be made to this compounds to improve efficacy.

Minor comment. 

HAART is an old terminology. The most popular is cART.

Author Response

The article by Bianco et al. presents a review and drug-like applicability of pyrrole-containing compounds that have potential to become anti-HIV compounds. The authors did a comprehensive analysis of these compounds and suggested that pyrrole ring-containing compounds can target multiple viral proteins including gp120 and RNase H domain of HIV-1 reverse transcriptase. In my opinion, it is an interesting review. However, the experimental data show that the compounds have high IC50 and it is not likely that these compounds will ever get to the clinical trials. The bar for developing anti-HIV compounds is very high. With the recent approval of cabotegravir and rilpivirine combination that acts a long-acting treatment, the focus of the anti-HIV drug development field is on the compounds that can be administered as long-acting compounds. That includes the new capsid inhibitor by GILEAD and Islatavir by MEERCK. Having said that, the door for new drugs is not necessarily closed. With this review presenting new perspectives, makes the field quite innovative and interesting. Thus, the review is worth publishing. However, there are some concerns that need to be addressed.

Point 1: The major concern is that it is unfair to call a compound pyrrole based when the pyrrole ring is fused to other ring. For example, in RDV (remdesevir), it is fused with another ring. That is the reason in IUPAC nomenclature is named as 4-aminopyrrolo. Additionally, RDV is not necessarily a pyrrole based drug. Also, only one of the three examples in Fig. 1 qualify as being the pyrrole-based drug. The same stands true for two entry inhibitors shown in Figure 2. The authors should be cautious to call these drugs as pyrrole-based.

Response: Dear reviewer, we appreciate the comments and the well-done revision. We want to report that we made the change in paragraph 5, according to the observation about the nomenclature.

Point 2: More details about the binding of the two entry inhibitors should be provided. The authors may want to include the location of V2/V3 loops of gp120 and a famous phenylalanine (F43) of CD4 and BMS-626529 bind.

Response: Done

Point 3: RNase H inhibitors are a long shot as anti-HIV drugs. This is because it is not clear at what step of viral replication, these compounds can act. Most certainly, these compounds are not good enough to compete with DNA/DNA or DNA/RNA complex of HIV-1 RT. Besides, there is no obvious binding pocket for pyrrole containing compounds to the RNase H domain of HIV-1 RT. Authors should comment on this fact.

Response: Done. Dear reviewer, we would like to report that we expanded the section 3 introduction, as shown in paragraphs 6, 7, and 8 of this one. In the discussion, we highlighted the function of the RNase H domain and why it is challenging to obtain compounds able to inhibit this function selectively.

Point 4: Only compound 4 has moderate IC50. The authors are suggested to comment on the IC50 of these compounds and suggest the modifications that can be made to this compounds to improve efficacy.

Response: Done.

Point 5: HAART is an old terminology. The most popular is cART.

Response: Done.

Reviewer 4 Report

The manuscript by Bianco et al. is a review that provides an update on pyrrole-based anti-HIV derivatives reported in literature between 2015 and 2020. Overall, the topic is interesting and the review is usefull. Authors did a fairly good job in summarizing the most relevant compounds related to the selected topic, but the review could benefit from the following considerations in order to make it more clear:

General comments

  • The description of the biological activities and the characterization of the mechanism of action of the reported compounds are not always clear. In some cases, info are divided between Figures and tables and not all the info reported in the figures are then discussed in the text. I think that this can generate confusion in the reader.
  • In the majority of cases, authors decided to present activity data in a table without structures and chemical structures in figures. This should be applied in all cases.
  • Chemical structures of the compounds were reported with different size along the manuscript. I would suggest to re-draw and make them all with same scale. Same thing for the text below each one.

Specific comments

  • In the abstract as well as in the introduction paragraph authors refer to HAART to define the antiretroviral therapy used to treat HIV infection. This definition is quite old, the terminology has evolved and replaced with the acronym cART, which stands for "combination antiretroviral therapy”.
  • Abstract, line six: suggest replacing “based on several chemical groups..” with “based on several chemical scaffolds”
  • Figure 1: Even if Figure 1 is nice, I found it not correct. The concept should be the presentation of some examples of approved drugs containing pyrrole moiety, since pyrrole is in common with all. Thus, why has remdesivir been placed outside with a particular arrow? Maybe the pyrrole can be placed in the center and the four drugs around it.
  • Figure 2: to be consistent with the other figures, numbers (1) and (2) should be inserted after compound names fostemsavir and temsavir, respectively.
  • Page 3 last paragraph: residue Asn428 is commented in the text but not reported in Figure 3. Instead, in the Figure 3 residue Asn432 is reported twice for two different residues and in the text the residue 432 is reported as Gln.
  • The chemical structure of compound 1 is reported in Figures 4, 5, and 6 together with Table 1. In Figure 4 it is underlined it is a racemic mixture while in table 1 the absolute configuration is reported. None info on absolute configuration for structures in Figures 5 and 6. Authors should clarify better.
  • The chemical structures of compounds 5, 6, and 7 are wrong. The CH2NH2 moiety in region III has been with the sole NH2.
  • The 1H-pyrrole-3-carbothioamide series are an important class to comment in this review. However, authors should stress a little bit more that even the more potent compounds identified within this class lack the ability to inhibit HIV replication.
  • Page 4 paragraph “Another important discovery was the dual activity of 1, as it inhibits both CCR5- and CXCR4-tropic HIV-1 with similar potency (IC50 of ~ 1.7-2.4 µM). In addition, 1 acts as a CD4-dependent viral inhibitor in a dose-dependent manner, exhibiting an IC50 of 2.4±0.2 µM. This compound competitively inhibits the CD4 receptor, blocking CD4-gp120 binding. Another interesting aspect is that 1 is also capable of inhibiting HIV-1 reverse transcriptase (RT) (IC50 of 43.4 µM).” Authors should add the literature reference and add a comment about the mechanism of action of compound 1. This compound is active in inhibiting HIV replication in the low micromolar range (about 2 uM). Its ability to inhibit RT activity is around 43 uM.   
  • Page 5: in the paragraph describing compounds 3 and 4 authors do not mentioned that this compounds were also co-crystallized with gp120. I would suggest to add this info for more clarity.
  • Page 6: Authors reported that starting from compound 1 further analogs were designed. However, in the description of chemical modifications made to obtain compound 5, the compound is compared to compound 4.
  • Page 7, first paragraph: “Additionally, the in vitro absorption, distribution, metabolism, excretion, and toxicity (ADMET) profiles of these compounds were similar to the profile of temsavir [35]”. I would suggest to add some more comments on the ADMET studies carried out and the results obtain.
  • Page 7, second paragraph: “Through an approach that can enhance interactions with gp120, a change in the CH2OH group from C-5 to C-4 in the thiazole moiety was proposed.” What kind of approach are authors referring to?
  • Figure 7: data about metabolic stability of compound 8 are reported in the figure even if these data were not discussed in the manuscript. Considering the % values indicated in the sentence “higher metabolic stability (93.5%)…”, I would suggest author to add a small footnote describing the meaning of those data, i.e. % of substrate remaining after…. incubation with….
  • Page 9: “However, guanidine analog 12 showed an excellent cytotoxicity profile (CC50 of 145.9 µM) compared with control molecule 4 (CC50 of 40.5 µM) (Table 3) and improved water solubility.” In this sentence authors should indicate that they are referring to predicted water solubility if they are considering the data reported in table 3 that were calculated using ChemAxon Software.
  • Page 11, last paragraph: authors should add a comment about cytotoxicity and SI for compounds 16 and 17.
  • Page 12: authors should indicate if the activity reported for the inhibition of RT-associated RNase H activity is referred to HIV-1 or HIV-2 RNase H or both.
  • Page 14: to be consistent along the manuscript all the activity should be converted in uM and not ug/mL.
  • Figure 14: the activities reported for compound 21-23 in the text are discussed as inhibition of HIV replication while in the figure they are reported as RT-inhibition. Which one is it correct?
  • Table 4 and Figure 16: which strain has been tested for those compounds? And in which cell line? I think “antiviral activity” is a too general definition for a review on anti-HIV compounds.
  • Page 16: what about the cytotoxicity of compound 34?

Author Response

Dear Reviewer,

Here we are sending the revision requested for the Manuscript ID: pharmaceuticals-1231782 for publication in Pharmaceutical special issue “Recent Developments in the Medicinal Chemistry of Pyrroles”.

Bellow we are providing a point-by-point response to the comments made by you as followed:

  • Comments from Reviewer

Point 1: The description of the biological activities and the characterization of the mechanism of action of the reported compounds are not always clear. In some cases, info are divided between Figures and tables and not all the info reported in the figures are then discussed in the text. I think that this can generate confusion in the reader. In the majority of cases, authors decided to present activity data in a table without structures and chemical structures in figures. This should be applied in all cases. Chemical structures of the compounds were reported with different size along the manuscript. I would suggest to re-draw and make them all with same scale. Same thing for the text below each one.

Response:  Dear Sir, we appreciate the comments and the well-done revision. We want to report that we changed all figures, making them more transparent for the reader. Besides, the font, size of the figures and the chemical structures have also been revised and modified as needed. We also standardize the format of the tables with no chemical structures. When possible, we consolidated data from different tables into a single table, such as data from table 2 were grouped in table 1, as well as data in table 5 were grouped with data in table 4 from the original manuscript. We hope that those changes made the manuscript more organized and easier for the comprehension of the readers.

Point 2: In the abstract as well as in the introduction paragraph authors refer to HAART to define the antiretroviral therapy used to treat HIV infection. This definition is quite old, the terminology has evolved and replaced with the acronym cART, which stands for "combination antiretroviral therapy”.

Response: Done.

Point 3: Abstract, line six: suggest replacing “based on several chemical groups..” with “based on several chemical scaffolds”

Response: Done.

Point 4: Figure 1: Even if Figure 1 is nice, I found it not correct. The concept should be the presentation of some examples of approved drugs containing pyrrole moiety, since pyrrole is in common with all. Thus, why has remdesivir been placed outside with a particular arrow? Maybe the pyrrole can be placed in the center and the four drugs around it.

Response: Done.

Point 5: Figure 2: to be consistent with the other figures, numbers (1) and (2) should be inserted after compound names fostemsavir and temsavir, respectively.

Response: Dear reviewer, we decided to standardize drugs already on the market with their commercial names, without numbers.

Point 6: Page 3 last paragraph: residue Asn428 is commented in the text but not reported in Figure 3. Instead, in the Figure 3 residue Asn432 is reported twice for two different residues and in the text the residue 432 is reported as Gln.

Response: Done.

Point 7: The chemical structure of compound 1 is reported in Figures 4, 5, and 6 together with Table 1. In Figure 4 it is underlined it is a racemic mixture while in table 1 the absolute configuration is reported. None info on absolute configuration for structures in Figures 5 and 6. Authors should clarify better.

Response: Done.

Point 8: The chemical structures of compounds 5, 6, and 7 are wrong. The CH2NH2 moiety in region III has been with the sole NH2.

Response: Done.

Point 9: The 1H-pyrrole-3-carbothioamide series are an important class to comment in this review. However, authors should stress a little bit more that even the more potent compounds identified within this class lack the ability to inhibit HIV replication.

Response: Done.

Point 10: Page 4 paragraph “Another important discovery was the dual activity of 1, as it inhibits both CCR5- and CXCR4-tropic HIV-1 with similar potency (IC50 of ~ 1.7-2.4 µM). In addition, 1 acts as a CD4-dependent viral inhibitor in a dose-dependent manner, exhibiting an IC50 of 2.4±0.2 µM. This compound competitively inhibits the CD4 receptor, blocking CD4-gp120 binding. Another interesting aspect is that 1 is also capable of inhibiting HIV-1 reverse transcriptase (RT) (IC50 of 43.4 µM).” Authors should add the literature reference and add a comment about the mechanism of action of compound 1. This compound is active in inhibiting HIV replication in the low micromolar range (about 2 uM). Its ability to inhibit RT activity is around 43 uM.

Response: Done.

Point 11:  Page 5: in the paragraph describing compounds 3 and 4 authors do not mentioned that this compounds were also co-crystallized with gp120. I would suggest to add this info for more clarity.

Response: Done.

Point 12: Page 6: Authors reported that starting from compound 1 further analogs were designed. However, in the description of chemical modifications made to obtain compound 5, the compound is compared to compound 4.

Response: Done.

Point 13: Page 7, first paragraph: “Additionally, the in vitro absorption, distribution, metabolism, excretion, and toxicity (ADMET) profiles of these compounds were similar to the profile of temsavir [35]”. I would suggest to add some more comments on the ADMET studies carried out and the results obtain.

Response: Done.

Point 14: Page 7, second paragraph: “Through an approach that can enhance interactions with gp120, a change in the CH2OH group from C-5 to C-4 in the thiazole moiety was proposed.” What kind of approach are authors referring to?

Response: Done.

Point 15:  Figure 7: data about metabolic stability of compound 8 are reported in the figure even if these data were not discussed in the manuscript. Considering the % values indicated in the sentence “higher metabolic stability (93.5%)…”, I would suggest author to add a small footnote describing the meaning of those data, i.e. % of substrate remaining after…. incubation with….

Response: Done.

Point 16: Page 9: “However, guanidine analog 12 showed an excellent cytotoxicity profile (CC50 of 145.9 µM) compared with control molecule 4 (CC50 of 40.5 µM) (Table 3) and improved water solubility.” In this sentence authors should indicate that they are referring to predicted water solubility if they are considering the data reported in table 3 that were calculated using ChemAxon Software.

Response: Done.

Point 17: Page 11, last paragraph: authors should add a comment about cytotoxicity and SI for compounds 16 and 17.

Response: Done.

Point 18: Page 12: authors should indicate if the activity reported for the inhibition of RT-associated RNase H activity is referred to HIV-1 or HIV-2 RNase H or both.

Response: Done.

Point 19: Page 14: to be consistent along the manuscript all the activity should be converted in uM and not ug/mL.

Response: Done.

Point 20: Figure 14: the activities reported for compound 21-23 in the text are discussed as inhibition of HIV replication while in the figure they are reported as RT-inhibition. Which one is it correct?

Response: Done.

Point 21: Table 4 and Figure 16: which strain has been tested for those compounds? And in which cell line? I think “antiviral activity” is a too general definition for a review on anti-HIV compounds.

Response: Done.

Point 22: Page 16: what about the cytotoxicity of compound 34?

Response: Dear reviewer, the authors do not mention the CC50 value for this compound.

Round 2

Reviewer 2 Report

This is a revised review article describing single molecule inhibitors of HIV-1 replication based on pyrrole chemistry.

The revised version is improved and the authors have responded to major concerns- first, better organizing the information in the figures and tables so that it will be easier for reviewers to navigate as a resource and, second, adding more description of the relevant chemistry and predicted mechanisms.

That said, several of the figures still include either extraneous information or insufficient details to understand exactly what the authors are trying to convey. For example-

  1. Figure 3 shows interactions that may or may not be interesting to readers, but what would be valuable would be to really explain how the drug is thought to "work" to block Envelope function.
  2. Figure 4 is ok but would be nice to see some indication in terms of mechanism as to how and why one of these is thought to be an agonist and the other is an antagonist.
  3. Figure 5 is hard to see with the blue background, and the text is not useful- basically what is said on the right is said on the left- so not really clear what they authors are trying to communicate with the arrows.
  4. Figure 6 is good showing both how pyrroles are scaffolds and some modifications that changed the activities, but the blue box with text is just reiterating what is already in the text of the review.
  5. Similarly, Figure 7 is just the same structure in Figure 6 with some text that is already in the text of the review.
  6. Figures 8 and 9 are fine but why not just summarize ALL of the permutations in a single figure? What is the point of having multiple?
  7. Similar issues for figures 10-18 as discussed above.

Minor-

1. Still several issues with English grammar and spelling throughout, e.g., this sentence is still not quite right- “..by 2019 retrovirus infected 38 million..". I think what is meant is, “As of 2019, there are 38 million people living with HIV infection worldwide.”

Author Response

Dear reviewer,

Here we are sending the revision requested for the Manuscript ID: pharmaceuticals-1231782 for publication in Pharmaceutical special issue “Recent Developments in the Medicinal Chemistry of Pyrroles”.

Bellow we are providing a point-by-point response to the comments made by you as followed:

1)         Comments from reviewer

Figure 3 shows interactions that may or may not be interesting to readers, but what would be valuable would be to really explain how the drug is thought to "work" to block Envelope function.

Response: Dear reviewer, the explanation about how temsavir “works” to block the Envelope function is described in section 2, paragraph 3. Figure 3 only has the purpose to inform about the importance of chemical groups in the temsavir’s activity.

2)         Comments from reviewer

Figure 4 is ok but would be nice to see some indication in terms of mechanism as to how and why one of these is thought to be an agonist and the other is an antagonist.

Response: Dear reviewer, the indication about these properties is in section 2, paragraph 6, according to the original paper.

3)         Comments from reviewer

Figure 5 is hard to see with the blue background, and the text is not useful- basically what is said on the right is said on the left- so not really clear what they authors are trying to communicate with the arrows.

Response: Done.

4)         Comments from reviewer

Figure 6 is good showing both how pyrroles are scaffolds and some modifications that changed the activities, but the blue box with text is just reiterating what is already in the text of the review.

Response: Done.

5)         Comments from reviewer

Similarly, Figure 7 is just the same structure in Figure 6 with some text that is already in the text of the review.

Response: Dear Sir, we decided to remove this figure once what we were trying to communicate is already in the text.

6)         Comments from reviewer

Figures 8 and 9 are fine but why not just summarize ALL of the permutations in a single figure? What is the point of having multiple?

Response: Done.

7)         Comments from reviewer

Similar issues for figures 10-18 as discussed above.

Response: Done.

7)         Comments from reviewer

Still several issues with English grammar and spelling throughout, e.g., this sentence is still not quite right- “..by 2019 retrovirus infected 38 million..". I think what is meant is, “As of 2019, there are 38 million people living with HIV infection worldwide.”

Response: Done.

Reviewer 4 Report

Page 7, There are 2 sentences about "ADME assessment" expressing the same concept. Please choice only one.

Author Response

Dear reviewer,

Here we are sending the revision requested for the Manuscript ID: pharmaceuticals-1231782 for publication in Pharmaceutical special issue “Recent Developments in the Medicinal Chemistry of Pyrroles”.

Bellow we are providing a point-by-point response to the comments made by you as followed:

1)         Comments from reviewer

Page 7. There are 2 sentences about "ADME assessment" expressing the same concept. Please choice only one.

Response: Done.